# BRAF V600E and Non-V600E Mutations in RAS Wild-Type Metastatic Colorectal Cancer: Prognostic and Therapeutic Insights from a Nationwide, Multicenter, Observational Study (J-BROS)

**DOI:** 10.3390/cancers17030399

**Published:** 2025-01-25

**Authors:** Hiroya Taniguchi, Kay Uehara, Toshiaki Ishikawa, Osamu Okochi, Naoya Akazawa, Hiroyuki Okuda, Hiroko Hasegawa, Manabu Shiozawa, Masato Kataoka, Hironaga Satake, Takaya Shimura, Chihiro Kondoh, Hidekazu Kuramochi, Toshihiko Matsumoto, Naoki Takegawa, Toshifumi Yamaguchi, Michitaka Nagase, Masato Nakamura, Nao Takano, Hideto Fujita, Takanori Watanabe, Tomohiro Nishina, Yasuhiro Sakamoto, Toshikazu Moriwaki, Hisatsugu Ohori, Masayoshi Nakanishi, Yosuke Kito, Setsuo Utsunomiya, Takeshi Ishikawa, Dai Manaka, Hiroshi Matsuoka, Takeshi Suto, Toshiyuki Arai, Shinichiro Shinzaki, Tohru Funakoshi, Goro Nakayama, Yuji Negoro, Yasushi Tsuji, Akitaka Makiyama, Kunio Takuma, Atsuki Arimoto, Katsunori Shinozaki, Ayako Mishima, Toshiki Masuishi

**Affiliations:** 1Department of Clinical Oncology, Aichi Cancer Center Hospital, Nagoya 464-8681, Japan; 2Department of Gastroenterological Surgery, Nagoya University Hospital, Nagoya 466-8560, Japan; 3Department of Gastroenterological Surgery, Nippon Medical School, Tokyo 113-8603, Japan; 4Department of Specialized Surgeries, Institute of Science Tokyo, Tokyo 113-8519, Japan; 5Department of Medical Oncology, Juntendo University, Tokyo 113-8431, Japan; 6Department of Surgery, Tosei General Hospital, Seto 489-8642, Japan; 7Department of Gastroenterological Surgery, Sendai City Medical Center Sendai Open Hospital, Sendai 983-0824, Japan; 8Department of Clinical Oncology, Keiyukai Sapporo Hospital, Sapporo 003-0026, Japan; 9Department of Gastroenterology and Hepatology, NHO Osaka National Hospital, Osaka 540-0006, Japan; 10Department of Gastrointestinal Surgery, Kanagawa Cancer Center, Yokohama 241-0815, Japan; 11Department of Surgery, NHO Nagoya Medical Center, Nagoya 460-0001, Japan; 12Department of Medical Oncology, Kobe City Medical Center General Hospital, Kobe 650-0047, Japan; 13Department of Medical Oncology, Kochi Medical School, Nankoku 783-8505, Japan; 14Department of Gastroenterology and Metabolism, Nagoya City University Graduate School of Medical Sciences, Nagoya 467-8602, Japan; 15Department of Medical Oncology, Toranomon Hospital, Tokyo 105-8470, Japan; 16Department of Medical Oncology, National Cancer Center Hospital East, Kashiwa 277-8577, Japan; 17Department of Chemotherapy, Tokyo Women’s Medical University Yachiyo Medical Center, Yachiyo 276-8524, Japan; 18Department of Medical Oncology, NTT Medical Center Tokyo, Tokyo 141-8625, Japan; 19Department of Internal medicine, Himeji Red Cross Hospital, Himeji 670-8540, Japan; 20Department of Medical Oncology, Ichinomiyanishi Hospital, Ichinomiya 494-0001, Japan; 21Department of Gastroenterology, Hyogo Cancer Center, Akashi 673-8558, Japan; 22Cancer Chemotherapy Center, Osaka Medical and Pharmaceutical University Hospital, Takatsuki 569-0801, Japan; 23Department of Medical Oncology, Saku Central Hospital Advanced Care Center, Saku 385-0051, Japan; 24Aizawa Comprehensive Cancer Center, Aizawa Hospital, Matsumoto 390-8510, Japan; 25Department of Surgery, Tokai Central Hospital, Kagamihara 504-8601, Japan; 26Department of Advanced Medicine, Nagoya University Hospital, Nagoya 466-8560, Japan; 27Department of General and Digestive surgery, Kanazawa Medical University, Uchinadamachi 920-0293, Japan; 28Department of Surgery, Japanese Red Cross Society Himeji Hospital, Himeji 670-8540, Japan; 29Department of Surgery, Tokushima Municipal Hospital, Tokushima 770-0812, Japan; 30Department of Gastrointestinal Medical Oncology, NHO Shikoku Cancer Center, Matsuyama 791-0245, Japan; 31Department of Medical Oncology, Osaki Citizen Hospital, Osaki 989-6183, Japan; 32Department of Gastroenterology, University of Tsukuba, Tsukuba 305-8576, Japan; 33Department of Gastroenterology and Hepatology, Kurashiki Central Hospital, Kurashiki 710-8602, Japan; 34Department of Medical Oncology, Ishinomaki Red Cross Hospital, Ishinomaki 986-8522, Japan; 35Department of Digestive Surgery, Kyoto Prefectural University of Medicine, Kyoto 602-8566, Japan; 36Department of Surgery, Matsushita Memorial Hospital, Moriguchi 570-8540, Japan; 37Department of Medical Oncology, Ishikawa Prefectural Central Hospital, Kanazawa 920-8530, Japan; 38Department of Clinical Oncology, Kainan Hospital, Yatomi 498-8502, Japan; 39Outpatient Oncology Unit, Department of Molecular Gastroenterology and Hepatology, Kyoto Prefectural University of Medicine, Kyoto 602-8566, Japan; 40Department of Surgery, Kyoto Katsura Hospital, Kyoto 615-8256, Japan; 41Department of Surgery, Fujita Health University, Toyoake 470-1192, Japan; 42Department of Gastroenterological Surgery, Yamagata Prefectural Central Hospital, Yamagata 990-2292, Japan; 43Department of Surgery, Anjo Kosei Hospital, Anjo 446-8602, Japan; 44Department of Gastroenterology, School of Medicine, Hyogo Medical University, Nishinomiya 663-8501, Japan; 45Department of Surgery, Asahikawa Kosei General Hospital, Asahikawa 078-8211, Japan; 46Department of Gastroenterological Surgery, Nagoya University Graduate School of Medicine, Nagoya 466-8560, Japan; 47Department of Oncological Medicine, Kochi Health Sciences Center, Kochi 781-8555, Japan; 48Department of Medical Oncology, Tonan Hospital, Sapporo 060-0004, Japan; 49Department of Hematology/Oncology, Japan Community Healthcare Organization Kyushu Hospital, Kitakyushyu 806-0034, Japan; 50Cancer Center, Gifu University Hospital, Gifu 501-1194, Japan; 51Department of Surgery, Tokyo Metropolitan Tama Medical Center, Fuchu 183-8524, Japan; 52Department of General Surgery, Toyohashi Municipal Hospital, Toyohashi 441-8570, Japan; 53Division of Clinical Oncology, Hiroshima Prefectural Hospital, Hiroshima 734-8530, Japan

**Keywords:** BRAF mutations, metastatic colorectal cancer, V600 mutation, non-V600 mutation, microsatellite instability-high, RAS wild-type, prognosis, chemotherapy regimens, bevacizumab, anti-EGFR therapy

## Abstract

In colorectal cancer treatment, identifying genomic abnormalities and selecting the appropriate treatment are crucial. In this prospective observational study, we analyzed 511 patients for RAS and BRAF mutations. Our key findings show that BRAF V600E and non-V600E mutations have different clinical and prognostic outcomes. However, the choice of first-line chemotherapy did not significantly affect treatment effectiveness in patients with BRAF mutations or in those with BRAF wild-type tumors in this real-world setting. These results provide valuable insights into treatment strategies and may help guide future research in colorectal cancer therapy.

## 1. Introduction

Colorectal cancer (CRC) is the third most commonly diagnosed cancer and the second leading cause of cancer-related deaths globally and in the United States [1,2]. In Japan, it is the most frequently diagnosed cancer and ranks second in cancer mortality [3]. In resectable cases, curative treatment is possible even with distant metastases [4]. For unresectable metastatic CRC (mCRC), systemic therapy remains the standard of care, with gradual improvements in outcomes leading to a median overall survival (OS) of approximately 30 months [5,6,7]. This progress has been driven by the introduction of novel cytotoxic agents, such as oxaliplatin, irinotecan, and trifluridine/tipiracil [8]. Moreover, targeted therapies, including monoclonal antibodies against vascular endothelial growth factors like bevacizumab (BEV) and those targeting epidermal growth factor receptor (EGFR), have demonstrated significant efficacy in mCRC [9,10].

Precision medicine has recently gained prominence in oncology, particularly for cancers like CRC, where genetic and molecular heterogeneity significantly influence disease progression and treatment response [11]. Among the most well-defined genomic alterations in CRC are KRAS mutations, occurring in 30–40% of cases. These mutations, predominantly in codons 12 and 13, lead to persistent activation of RAS proteins by preventing GTP hydrolysis, thereby maintaining downstream signaling and driving oncogenic activity independently of external growth signals [12]. Clinically, KRAS mutations are a known predictor of resistance to anti-EGFR therapies such as cetuximab and panitumumab, as mutated RAS circumvents EGFR blockade [13,14]. Similarly, NRAS mutations, which occur in 2–5% of CRC cases, involve less common mutations in codons 61 and 146 and are linked to anti-EGFR treatments [15]. These findings underscore the importance of testing for RAS (KRAS/NRAS) mutations in clinical practice.

BRAF mutations are present in approximately 8–10% of CRC cases and generally occur independently of RAS mutations. The most common alteration is the V600E mutation, which constitutes the majority of BRAF mutations in CRC [16]. BRAF, a serine/threonine kinase in the downstream RAS signaling pathway, becomes constitutively active due to the V600E mutation, promoting oncogenic signaling even without upstream RAS activation [17]. Clinically, BRAF V600E mutations are linked to unique features, such as poor prognosis, a higher prevalence of microsatellite instability-high (MSI-H) and a propensity for metastatic spread, particularly to the peritoneum [18]. Unlike KRAS and NRAS mutations, the benefit of anti-EGFR monoclonal antibody therapies in BRAF-mutated CRC remains uncertain [19,20]. Given the aggressive nature of BRAF-mutated mCRC, there is a pressing need to explore more intensive or novel therapeutic strategies for these patients.

Although BRAF V600E mutations are associated with aggressive disease, non-V600E mutations are less common but may have important implications for prognosis and treatment. Early studies suggest that non-V600E BRAF mutations are linked to a more indolent disease course, reduced metastatic potential, and better OS compared to V600E mutations [21,22]. The role of anti-EGFR therapy in these cases remains uncertain [23,24]. Moreover, non-V600E mutations may respond differently to targeted treatments, such as BRAF and MEK inhibitors, which have demonstrated efficacy in BRAF V600E mutations [25,26]. This highlights the need to consider non-V600E mutations in personalized treatment strategies. However, their rarity means that optimal therapeutic approaches for non-V600E BRAF-mutant CRC are not yet well defined, emphasizing the importance of ongoing research to clarify their prognostic and predictive value.

This multicenter prospective observational study aims to examine the prevalence of BRAF mutations in RAS wild-type mCRC and explore the clinical and pathological features, prognosis, and the impact of treatment regimens based on BRAF status in real-world settings. This study seeks to provide valuable insights into the molecular profiles of CRC, which could inform future treatment strategies and enhance outcomes for mCRC patients.

## 2. Materials and Methods

### 2.1. Patients

This prospective observational study included patients with histologically confirmed unresectable colorectal adenocarcinoma. At the time of consent, patients had either already begun or were scheduled to begin first-line systemic chemotherapy. Patients with concurrent malignancies significantly affecting prognosis or those with known RAS or BRAF mutations were excluded. Tumor and lymph node staging followed the seventh edition of the TNM Classification of Malignant Tumors. Histologically, tumors were classified into well or moderately differentiated adenocarcinoma (tub1 and tub2) and others, such as poorly differentiated adenocarcinoma, mucinous adenocarcinoma, and signet-ring cell carcinoma (por/muc/sig) [27]. This study adhered to the ethical guidelines of the Declaration of Helsinki, and the protocol was approved by the Institutional Review Boards of participating institutions, including Aichi Cancer Center (registry number: 2015-1-185). This study was registered in the UMIN Clinical Trials Registry before commencement (UMIN000021002). All participants provided written informed consent before participation.

### 2.2. RAS, BRAF and MSI Testing

In this study, RAS and BRAF mutations were centrally assessed at a laboratory (G&G Science Co., Ltd., Fukushima, Japan), with results provided to the investigator and study office within 2 weeks. Genomic DNA was extracted from formalin-fixed, paraffin-embedded tissue samples collected from surgical resections or biopsies of CRC patients. Testing was performed following the manufacturer’s protocol using the Luminex xMAP bead-based multiplex immunoassay system (Luminex). KRAS and NRAS mutations in codons 12, 13, 59, 61, 117, and 146, as well as BRAF V600E, were identified using the PCR-based MEBGEN RASKET-B Kit (Medical & Biological Laboratories Co., Tokyo, Japan) [28].

We also used the Genosearch™ BRAF kit, which detects mutations in BRAF V600 (including V600E, V600K, V600D, and V600R) as well as mutations outside V600, such as exon 11 mutations in codons 464 (e.g., G464E, G464V, G464R), 466 (G466R, G466V, G466E), 467 (S467L), 469 (G469A, G469V, G469R, G469E), and 485 (L485F). The kit was also used to identify mutations in codons 524 (Q524L), 525 (L525R), 581 (N581S, N581I, N581T), 594 (D594N, D594G), 596 (D596R), 597 (L597R, L597S, L597V, L597Q, L597P), 598 (A598T), 599 (T599_600insT), and 601 (V601E, V601N) [29].

Additionally, the MSI status was determined using PCR-based testing with the quasi-monomorphic variation range method [30]. MSI testing was conducted in tumors with BRAF mutations only. The MSI status was classified as MSI-H if two or more markers were unstable, MSI-low if one marker was unstable, and microsatellite stable if no markers were unstable.

### 2.3. Treatment and Efficacy Assessment

Patient characteristics were recorded at the start of first-line chemotherapy. Clinical data were collected from RAS wild-type patients during a 36-month follow-up period. Right-sided colon cancer was defined as cancer of the cecum and ascending colon up to the splenic flexure. The choice of treatment regimen was determined by each investigator. In this study, a regimen combining fluoropyrimidine with either oxaliplatin or irinotecan was considered a doublet, while a regimen including all three agents was classified as a triplet. Tumor assessments to evaluate efficacy were conducted approximately every 8–10 weeks. Each investigator assessed tumor response based on RECIST (RECIST v1.1 criteria) v1.1 criteria for patients with measurable lesions, with no central review of response data.

### 2.4. Sample Size and Statistical Analysis

The sample size was calculated based on previous reports indicating that BRAF V600E and non-V600E mutations occur in about 5% of RAS wild-type cases. A target sample size of 1000 was set to ensure adequate precision, with the confidence interval (CI) width kept around 4%. Fisher’s exact test and a trend test were used to assess the associations between BRAF status and clinicopathological factors. OS was defined as the time from treatment initiation to death from any cause, while progression-free survival (PFS) was defined as the time from treatment initiation to either disease progression or death from any cause. OS and PFS were analyzed using Kaplan–Meier curves, and the log-rank test and hazard ratios (HRs) were calculated using the Cox proportional hazards model with 95% CIs. Statistical analysis was performed with EZR (Saitama Medical Center, Jichi Medical University, Saitama, Japan), a user interface for R (The R Foundation for Statistical Computing, Vienna, Austria). All tests were two-sided, with *p*-values < 0.05 considered statistically significant.

## 3. Results

### 3.1. Patient Characteristics and Prognosis

The occurrence of BRAF non-V600E mutations was lower than anticipated, but enough BRAF V600E cases were collected, leading to early termination of enrollment. Between January 2017 and March 2019, 511 patients were enrolled from 32 centers in Japan, with 377 RAS wild-type patients included in this analysis. The data cutoff date was 31 March 2022. The BRAF mutation rate was 21.0% (79/377), consisting of 71 (89.9%) BRAF V600E mutations and 8 (10.1%) BRAF non-V600E mutations. Additional MSI testing in patients with BRAF mutations showed an MSI-H rate of 11.3% (8/71), all of which were in the BRAF V600E group.

Patient and disease characteristics are summarized in Table 1. The median age of all patients was 66 years, with 64% being male and 72% having left-sided tumors. Compared to RAS/BRAF wild-type patients, the BRAF V600E group had a higher proportion of females, right-sided tumors and por/muc/sig histology. Metastatic lesions were similar between the two groups. Baseline serum carbohydrate antigen 19-9 (CA19-9) levels were higher in the BRAF V600E mutation group, while carcinoembryonic antigen (CEA) levels were comparable between the groups. The BRAF non-V600E-mutant group showed characteristics more similar to the RAS/BRAF wild-type patients than to the BRAF V600E group. When comparing MSI-H and non-MSI-H patients in the BRAF V600E group, MSI-H patients were older and had lower levels of CEA and CA19-9.

Regarding prognosis, the BRAF V600E-mutant group had significantly worse outcomes than the BRAF wild-type group (median OS 12.4 months vs. 37.5 months, HR 3.24, 95% CI 2.38–4.42, *p* < 0.001) (Figure 1a). In contrast, patients with BRAF non-V600E mutations had better outcomes than those with BRAF V600E mutations (median OS 34.7 months vs. 12.4 months, HR 0.61, 95% CI 0.37–1.01, *p* = 0.047) (Figure 1b), with no significant difference compared to the BRAF wild-type group. Moreover, in the BRAF V600E-mutant group, MSI-H patients had a better prognosis than non-MSI-H patients (median OS 12.4 months vs. 30.9 months, HR 0.37, 95% CI 0.14–0.97, *p* = 0.036).

### 3.2. Treatment Effectiveness

We then evaluated the efficacy of first-line treatment in the BRAF V600E and BRAF wild-type groups. Treatment details and distributions are provided in Table 2. Moreover, the overall response rate (ORR) was higher in the BRAF wild-type group compared to the BRAF-mutant group (66.3% vs. 43.8%, *p* < 0.001). A similar trend was observed for disease control rate (93.2% vs. 70.3%, *p* < 0.001).

In the BRAF V600E-mutant group, 45 patients received a doublet regimen, and 14 received a triplet regimen. The ORR was similar between the two groups (doublet vs. triplet, 47.7% (21/44) vs. 42.9% (6/14), *p* = 0.75). Regarding prognosis, PFS and OS were comparable (median PFS doublet vs. triplet, 6.44 months vs 8.28 months, HR 1.04, 95% CI 0.54–2.02, *p* = 0.90, Figure 2a; median OS, 12.2 months vs. 13.0 months, HR 1.18, 95% CI 0.62–2.23, *p* = 0.61, Figure 2b). Next, we explored the impact of anti-biologics on treatment effectiveness. Patients receiving BEV and anti-EGFR had median PFS of 8.8 and 5.7 months, respectively (HR 1.76, 95% CI 0.97–3.20, *p* = 0.060) (Figure 2c), suggesting a trend toward poorer outcomes in the anti-EGFR group, although not statistically significant. Median OS was 12.8 months in the BEV group and 13.8 months in the anti-EGFR group (HR 1.28, 95% CI 0.70–2.33; *p* = 0.42) (Figure 2d). The ORR was 51.9% (14/27) for BEV and 45.8% (11/24) for anti-EGFR.

In the BRAF non-V600E group, only seven patients were available for efficacy analysis, with three showing confirmed response. These patients were treated with FOLFOX plus BEV, FOLFOX plus panitumumab, or capecitabine alone (Table 3).

In the BRAF wild-type group, 219 (78.5%) patients received a doublet regimen, 23 (8.2%) received monotherapy, and 37 (13.3%) received a triplet regimen. Additionally, 23 (8.3%) received chemotherapy alone, 129 (46.6%) received a BEV combination, and 125 (45.1%) received an anti-EGFR combination. To assess the effectiveness of doublet plus BEV versus doublet plus EGFR in RAS and BRAF wild-type patients, a comparison was made between these two groups. In terms of tumor response, the ORR was similar for the doublet plus BEV (*n* = 84) and doublet plus EGFR (*n* = 111) groups, with 67.9% (57/84) and 68.4% (76/111), respectively. Regarding prognosis, PFS was comparable in both groups (median PFS, BEV vs. EGFR, 11.2 months vs. 11.0 months, HR 1.18, 95% CI 0.86–1.62, *p* = 0.30) (Figure 3a), and OS was also similar (34.5 months vs. 35.7 months, HR 1.20, 95% CI 0.84–1.71, *p* = 0.33) (Figure 3b). Finally, considering reports suggesting that the primary tumor location affects the efficacy of anti-EGFR antibodies, the relationship between primary tumor location and treatment efficacy was evaluated. However, even when analysis was restricted to left-sided tumors, no significant differences in PFS or OS were found between the BEV and EGFR groups (PFS, 11.2 vs. 11.1 months, HR 1.10, 95% CI 0.77–1.56, *p* = 0.61; OS, 37.5 vs. 35.7 months, HR 1.22, 95% CI 0.81–1.83, *p* = 0.33) (Figure 3c,d).

## 4. Discussion

In this multicenter, prospective observational study, BRAF V600E mutations were found to be relatively common and linked to a poor prognosis, while BRAF non-V600E mutations were less frequent than anticipated and associated with a better prognosis in RAS wild-type mCRC. Additionally, the study prospectively assessed the relationship between BRAF status and treatment efficacy. In this real-world setting, no significant differences in treatment effectiveness were observed between doublet or triplet regimens, nor between the use of antibody drugs in patients with BRAF V600E mutations. Similarly, for patients with RAS/BRAF wild-type tumors, no major differences were found between the use of BEV or EGFR antibodies.

In this study, BRAF V600E mutations were found in 22% of RAS wild-type cases, which is higher than previously reported. One possible explanation for this higher rate is that, when the study began, BRAF V600E testing was not yet widely used in clinical practice in Japan. Participation in the study provided an opportunity for BRAF testing, which may have led investigators to preferentially enroll patients with characteristics indicative of BRAF V600E mutations, such as right-sided colon cancer or peritoneal dissemination.

The clinical features observed in patients with BRAF V600E mutations, such as a higher prevalence of right-sided colon origin, poorly differentiated histology, and frequent peritoneal dissemination, were consistent with previous studies [18,31]. Elevated CA19-9 levels were also observed, in line with existing literature [32]. While MSI status was also evaluated, the frequency of MSI-H was relatively low. No significant clinicopathological differences were found between MSI-H and microsatellite stable (MSS) cases with BRAF V600E mutations, although a trend for lower CEA and CA19-9 levels was noted, which has not been commonly reported. In this study, MSI-H cases had a better prognosis than MSS cases in BRAF V600E mutation, potentially reflecting the survival benefit of immune checkpoint inhibitors in later-line chemotherapy for MSI-H patients [33]. However, due to the small number of MSI-H cases, further research is needed.

In this study, various treatments were given to patients with BRAF V600E mutations, but no significant differences in efficacy were found. The comparison between doublet and triplet regimens showed no notable differences in response rate, PFS, or OS. Previous studies have yielded mixed results; a subgroup analysis from the TRIBE trial [34] suggested that triplet plus BEV therapy might improve prognosis compared to doublet plus BEV therapy in BRAF V600E mutants, while pooled and retrospective analyses found no significant difference between the regimens [35,36]. Despite being based on real-world data, our study showed no differences, indicating that the triplet regimen, associated with higher toxicity, may not be necessary as first-line treatment.

In comparison between BEV and EGFR combination therapies in BRAF V600E-mutant cases, PFS was slightly better with the BEV combination, though not statistically significant, and OS showed no difference. This aligns with the results of a recent randomized control trial (RCT) [37]. For pretreated BRAF V600E-mutant mCRC, EGFR inhibitor plus BRAF inhibitor therapy has become a standard treatment based on RCT findings [26,38]. While few patients in this study received BRAF inhibitor in second- or later-line therapy, it would be reasonable to consider BEV combination therapy as first-line treatment if EGFR inhibitor plus BRAF inhibitor therapy is planned for second-line treatment. Based on the findings of this study, it can be observed that such treatment selection is also being implemented in real-world clinical practice.

In this study, the observed frequency of 2.1% is within the previously reported range of 1.6–5.5% for BRAF non-V600E mutations [22,23,39,40]. This variation can be attributed to differences in sample size, the range of BRAF mutations targeted, patient populations, and detection methods used in previous studies. Although the 2.1% frequency reported here is on the lower end, it is also lower than the frequency we previously observed in a Japanese cohort using the same methodology [41]. This may be due to the current cohort’s higher proportion of colon cancer patients and those with peritoneal dissemination, where the frequency of BRAF non-V600E mutations is generally low.

Although the number of patients with non-V600E mutations was small (eight cases), our analysis indicated that their clinicopathological characteristics were more similar to those of BRAF wild-type patients than those with BRAF V600E mutations. While there was no significant difference in prognosis, patients with BRAF non-V600E mutations had a better prognosis than those with BRAF V600E mutations, which aligns with previous studies [22]. Due to the small sample size, the optimal treatment for BRAF non-V600E mutations remains unclear in this study. Recent studies have classified BRAF mutations into three classes based on their reliance on RAS signaling and ability to signal as a monomer [42,43], with some reports suggesting that class 3 mutations may respond better to anti-EGFR antibodies. In our cohort, focusing on first-line treatments, there were responses to FOLFOX plus panitumumab. Additionally, recent studies have highlighted the efficacy of BRAF and MEK inhibitors and prospective clinical trials are ongoing [44,45]. Determining the optimal treatment for BRAF non-V600E mutations will require further research.

This study included a large number of patients with RAS/BRAF wild-type status and while various treatments were used for these patients, no significant differences in treatment efficacy or survival outcomes were observed. For example, no differences were found between EGFR antibodies and BEV. Previous RCTs have reported mixed results; the FIRE-3 trial showed better OS with EGFR combination therapy, while the CALGB80405 trial found no difference [46,47]. Later, a pooled analysis of RCTs confirmed the predictive value of primary tumor location [48], and the recent Japanese PARADIGM trial prospectively showed improved OS with anti-EGFR therapy compared to BEV, particularly in left-sided tumors [49]. The difference between RCTs and this study, which used real-world data from a diverse patient population, may reflect that variability, similar to comparisons with other real-world data [50]. Additionally, since treatment regimens were chosen by each investigator, anti-EGFR antibodies may have been more commonly used in patients with higher tumor burdens or poorer prognoses, potentially influencing the observed outcomes. Reflecting on these results, anti-EGFR antibodies are generally selected as first-line treatment for left-sided tumors. However, in patients who do not meet the eligibility criteria for clinical trials, BEV may also be considered as a treatment option.

This study has several limitations. First, as a prospective observational study rather than an RCT, treatment regimens were chosen by each investigator, which may have influenced the results. Additionally, since genomic mutation testing results were available to investigators within about 2 weeks, treatment decisions may have been affected by this information, introducing potential bias. However, the inclusion of a diverse patient population makes these findings valuable as real-world data for clinical practice. Another limitation is that recently proven effective treatments, such as BRAF inhibitors and immune checkpoint inhibitors for MSI-H patients, were rarely used, so their impact could not be assessed. Additionally, BRAF non-V600E mutations were detected using PCR, which only identifies specific mutation types, leaving other non-V600E mutations undetected. Moreover, MSI testing was only by PCR-based assay, which cannot detect elevated microsatellite alterations at selected tetranucleotide repeats that may also have prognostic and predictive value. However, PCR is cost-effective and highly sensitive, which may be advantageous if widespread testing becomes more common in the future. Lastly, while personalized treatment for CRC has advanced rapidly, this study did not examine other genetic abnormalities, such as HER2 or MET, beyond BRAF. Conducting comprehensive analyses with NGS and tracking circulating tumor DNA over time could provide deeper insights into identifying the most effective treatment strategies. These approaches will form the next phase of our research.

## 5. Conclusions

In this multicenter, prospective, observational study, BRAF V600E and non-V600E mutations exhibited distinct clinical and prognostic differences, with V600E mutations associated with worse outcomes. However, in a real-world setting, the choice of first-line chemotherapy or the use of antibody drugs did not significantly affect survival for patients with BRAF mutations or RAS/BRAF wild-type tumors.

## Figures and Tables

**Figure 1 cancers-17-00399-f001:**
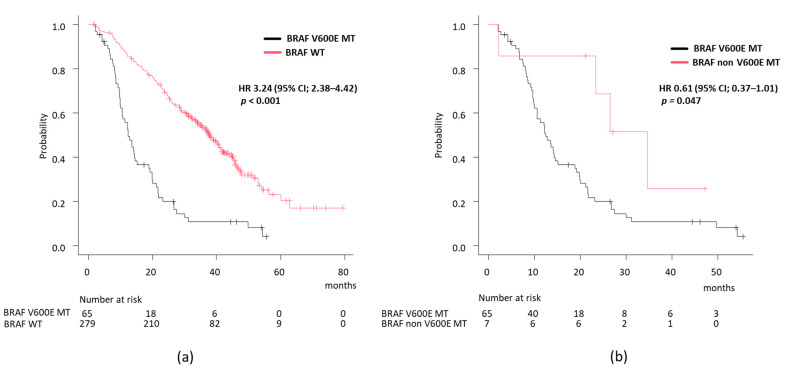
Overall survival according to BRAF status. (**a**) BRAF V600E-mutant group versus BRAF wild-type group; (**b**) BRAF V600E-mutant group versus BRAF non-V600E mutant group.

**Figure 2 cancers-17-00399-f002:**
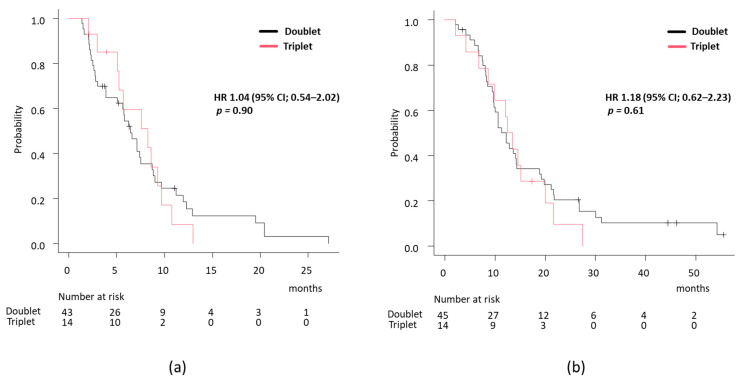
Survival analysis according to treatment regimen in BRAF V600E-mutant group. (**a**) PFS and (**b**) OS analysis between doublet versus triplet regimen.; (**c**) PFS and (**d**) OS analysis between BEV versus anti-EGFR group.

**Figure 3 cancers-17-00399-f003:**
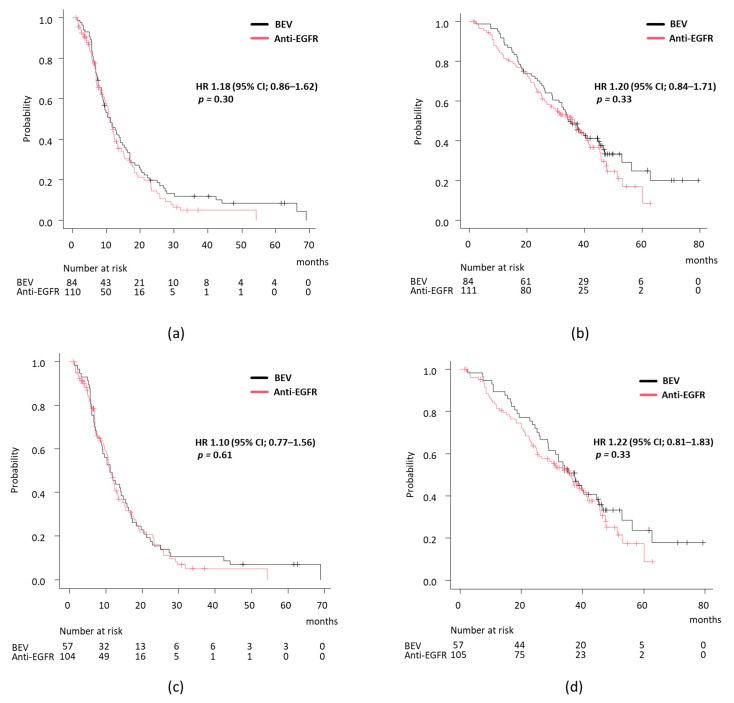
Survival analysis according to treatment regimen in BRAF wild-type group. (**a**) PFS and (**b**) OS analysis between BEV versus anti-EGFR group in whole population.; (**c**) PFS and (**d**) OS analysis between BEV versus anti-EGFR group in patients with left-sided tumors.

**Table 1 cancers-17-00399-t001:** Patient and disease characteristics.

		BRAF WT (N = 298)	BRAF V600E (N = 71)	BRAF NonV600E (N = 8)	BRAF V600E MSI-H (N = 8)	BRAF V600E Non MSI-H (N = 62) *
Age	Median	65.5	67.0	64.5	74	65.5
	Range	31–89	32–88	35–77	67–81	32–88
Sex	Male	201 (67.9)	35 (50.0)	3 (37.5)	4 (50.0)	31 (50.0)
	Female	95 (32.1)	35 (50.0)	5 (62.5)	4 (50.0)	31 (50.0)
ECOG PS	0	196 (66.2)	50 (71.4)	5 (62.5)	6 (75.0)	44 (71.0)
	1	92 (31.1)	15 (21.4)	2 (25.0)	0 (0.0)	15 (24.2)
	≥2	8 (2.7)	5 (7.1)	1 (12.5)	2 (25.0)	3 (4.8)
Primary site	Left	242 (81.8)	22 (31.4)	6 (75.0)	1 (12.5)	21 (33.9)
	Right	54 (18.2)	48 (68.6)	2 (25.0)	7 (87.5)	41 (66.1)
Histology	Tub 1/Tub 2	267 (90.2)	46 (66.7)	8 (100.0)	4 (50.0)	42 (68.9)
	Por/Muc/Sig	29 (9.8)	23 (33.3)	0 (0.0)	4 (50.0)	19 (31.1)
Metastatic Organ	Lymph node	93 (32.5)	32 (45.7)	3 (37.5)	6 (75.0)	26 (41.9)
	Liver	171 (59.8)	43 (61.4)	6 (75.0)	2 (25.0)	41 (66.1)
	Lung	84 (29.4)	14 (20.0)	3 (37.5)	0 (0.0)	14 (22.6)
	Peritoneum	70 (24.5)	23 (32.9)	2 (25.0)	1 (12.5)	22 (35.5)
CEA	Mean	448.2	365.5	188.8	7.6	413.2
	Median	15.3	11.3	42.4	3.2	21.0
CA 19-9	Mean	449.2	6056.9	121.3	47.7	6871.7
	Median	21.5	212.9	13.1	29.0	266.5

* One patient’s data are missing.

**Table 2 cancers-17-00399-t002:** Patient and disease characteristics for treatment effectiveness analysis.

		BRAF WT (N = 279)	BRAF V600E (N = 65)	BRAF NonV600E (N = 7)	BRAF V600E MSI-H (N = 8)	BRAF V600E Non MSI-H (N = 57)
Age	<70 years	182 (65.2)	43 (66.2)	4 (57.1)	1 (12.5)	42 (73.7)
	70 ≤ years	97 (34.8)	22 (33.8)	3 (42.9)	7 (87.5)	15 (26.3)
Sex	Male	191 (68.5)	33 (50.8)	3 (42.9)	4 (50.0)	29 (50.9)
	Female	88 (31.5)	32 (49.2)	4 (57.1)	4 (50.0)	28 (49.1)
ECOG PS	0	180 (64.5)	46 (70.8)	5 (71.4)	6 (75.0)	40 (70.2)
	1	91 (32.6)	14 (21.5)	2 (28.6)	0 (0.0)	14 (24.6)
	≥2	8 (2.9)	5 (7.7)	0 (0.0)	2 (25.0)	3 (5.3)
Primary site	Left	228 (81.7)	18 (27.7)	5 (71.4)	1 (12.5)	17 (29.8
	Right	51 (18.3)	47 (72.3)	2 (28.6)	7 (87.5)	40 (70.2)
Primary tumor	Present	75 (26.9)	20 (30.8)	1 (14.3)	2 (25.0)	18 (31.6)
	Absent	204 (73.1)	45 (69.2)	6 (85.7)	6 (75.0)	39 (68.4)
Metastatic Organ	Lymph node	90 (33.3)	32 (49.2)	2 (28.6)	6 (75.0)	26 (45.6)
	Liver	165 (61.1)	41 (63.1)	5 (71.4)	2 (25.0)	39 (68.4)
	Lung	82 (30.4)	14 (21.5)	2 (28.6)	0 (0.0)	14 (24.6)
	Peritoneum	64 (23.7)	21 (32.3)	1 (14.3)	1 (12.5)	20 (35.1)
Cytotoxic drugs	Mono	23 (8.2)	5 (7.8)	2 (28.6)	1 (14.3)	4 (7.0)
	Doublet	219 (78.5)	45 (70.3)	4 (57.1)	5 (71.4)	40 (70.2)
	Triplet	37 (13.3)	14 (21.9)	1 (14.3)	1 (14.3)	13 (22.8)
Anti-biologics	None	23 (8.3)	12 (19.0)	1 (14.3)	2 (28.6)	10 (17.9)
	BEV	129 (46.6)	27 (42.9)	2 (28.6)	2 (28.6)	25 (44.6)
	Anti-EGFR	125 (45.1)	24 (38.1)	4 (57.1)	3 (42.9)	21 (37.5)

**Table 3 cancers-17-00399-t003:** Case series of BRAF non-V600E mutant colorectal cancer.

No.	BRAF Mutation	Age	Sex	Primary Site	Regimen	Response	PFS	OS
63	BRAF D594G	35	Female	Sigmoid	FOLFOX + anti-EGFR	SD	4.8	21.2+
68	BRAF G466R	61	Male	Transverse	FOLFOX + BEV	PR	22.0	34.7
102	BRAF D594G	70	Male	Rectosigmoid	FOLFOX + anti-EGFR	PR	5.5+	47.2+
239	BRAF D594G	68	Male	Rectosigmoid	Capecitabine	PR	8.3	26.6
249	BRAF N581S	70	Female	Rectum	FOLFOXIRI + BEV	SD	6.4	23.4
298	BRAF D594G	47	Female	Transverse	FOLFOX + anti-EGFR	SD	1.9+	2.2
347	BRAF D594G	77	Female	Rectum	IRI + anti-EGFR	SD	7.4	27.2+

## Data Availability

Data are available within the article.

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
