# Peer review of "BRAF V600E and Non-V600E Mutations in RAS Wild-Type Metastatic Colorectal Cancer: Prognostic and Therapeutic Insights from a Nationwide, Multicenter, Observational Study (J-BROS)"

_cancers, 2025, doi:10.3390/cancers17030399_

Round 1
Reviewer 1 Report
Comments and Suggestions for Authors
The study is of interest to a wider scientific audience in the field of molecular medicine as well as of molecular oncology and oncologists.
The main findings are: BRAF V600E mutations were relatively common and linked to a poor prognosis. BRAF non-V600E mutations were less frequent and associated with a better prognosis in RAS wild-type mCRC. No significant differences in treatment effectiveness were observed between double or triplet regimens, nor between the use of antibody drugs in patients with BRAF V600E mutations. In patients with RAS/BRAF wild-type tumors, no major differences were found between the use of BEV or EGFR antibodies.
The title however does not entirely reveal the content of the study, therefore in my opinion perhaps more insightful title that would encompass the entire content of the paper and its significance both in research as well as clinics would be more appropriate.
The simple summary and abstract accurately reflect the content of the study. The introduction is well-written. The study has the appropriate ethical approvals and is registered. Mutations were genotyped at one place which is always good especially when archived FFPE samples are used. Only problem that I see is that only BRAF positive samples were tested for MSI. Although it is true that KRAS mutations are more frequent in stable tumors and BRAF in MSI-H tumors, the subset of tumors with EMAST type of instability lies somewhere in between and this is missed by classical MSI detection. And this subtype has some characteristics of stable and some of MSI tumors in terms of survival and response to therapy. Materials and methods are otherwise well presented. Results are clear and well presented, figures and tables are appropriate.
Discussion is well written and argumented. Perhaps the MSI part of discussion can take into account the MSI typing methods that do not reveal the samples with EMAST type of MSI. The results are discussed in the context of other studies appropriately and the literature is up to date.
Author Response
Thank you for pointing this out. We agree with this comment. Therefore, we added as limitation of this study in Discussion session (Line 407-410) as follows; Moreover, MSI testing was only by PCR-based assay, which cannot detect elevated microsatellite alterations at selected tetranucleotide repeats that may also have prognostic and predictive value.
And changed our manuscript title as follows: BRAF V600E and non-V600E mutations in RAS wild-type metastatic colorectal cancer: Prognostic and therapeutic insights from a nationwide, multicenter, observational study (J-BROS)
Reviewer 2 Report
Comments and Suggestions for Authors
1) General comments
Dr. Taniguchi, et al. investigated to “ BRAF V600E and non-V600E mutational status in Japanese patients with RAS wild-type mCRC: A nationwide, multicenter, observational study (J-BROS)”. This article is informative and well presented. The reviewer has some comments.
1. Please describe the further strategies or directions for this study to reveal other genetic abnormalities and treatments in Discussion.
Author Response
Comment 1: Please describe the further strategies or directions for this study to reveal other genetic abnormalities and treatments in Discussion.
Thank you for pointing this out. We agree with this comment. Therefore, we added the sentence in Discussion session (Line 413-416); Conducting comprehensive analyses with NGS and tracking circulating tumor DNA over time could provide deeper insights into identifying the most effective treatment strategies. These approaches will form the next phase of our research.
Reviewer 3 Report
Comments and Suggestions for Authors
BRAF V600E and non-V600E mutational status in Japanese patients with RAS wild-type mCRC: A nationwide, multicenter, observational study (J-BROS), examines the effects of BRAF mutation on treatment outcomes. While the target numbers for enrollment were met, the set of BRAF non-V600 seems too small to reach any real conclusion, despite the claim of statistical significance in Fig. 1B. I would like to see the 95% confidence interval drawn on that graph for both sets. Based on the discussion, it seems that the numbers achieved are lower than anticipated but could have been predicted from the range of frequencies in other studies and clinical presentation of the study population. Given the disappointing results for treatment outcomes, is this multicenter consortium considering changes in treatment based on statements on lines 371-378 and lines 420-424? Did this report have any immediate impact even within the participating members and their associates? I would like the have the p values included on the graphs for quicker reference.
Author Response
Comment 1: “I would like to see the 95% confidence interval drawn on that graph for both sets” and “I would like the have the p values included on the graphs for quicker reference.”
Thank you for pointing this out. We agree with this comment. Therefore, we added HR, 95%CI and p-value in Figures 1-3.
Comment 2: Given the disappointing results for treatment outcomes, is this multicenter consortium considering changes in treatment based on statements on lines 371-378 and lines 420-424? Did this report have any immediate impact even within the participating members and their associates?
Thank you for pointing this out. This study is designed with a long observation period, and the current treatment strategies are determined based not only on the results of this trial but also on the evidence that emerged during the observation period. Therefore, we added the sentences as follows: Based on the findings of this study, it can be observed that such treatment selection is also being implemented in real-world clinical practice (Line 357-359), and “Reflecting on these results, anti-EGFR antibodies are generally selected as first-line treatment for left-sided tumors. However, in patients who do not meet the eligibility criteria for clinical trials, BEV may also be considered as a treatment option (Line 395-397).